# *Bifidobacterium longum* LBUX23 Isolated from Feces of a Newborn; Potential Probiotic Properties and Genomic Characterization

**DOI:** 10.3390/microorganisms11071648

**Published:** 2023-06-24

**Authors:** Pedro A. Reyes-Castillo, Raquel González-Vázquez, Edgar Torres-Maravilla, Jessica I. Bautista-Hernández, Eduardo Zúñiga-León, Martha Leyte-Lugo, Leovigildo Mateos-Sánchez, Felipe Mendoza-Pérez, María Angélica Gutiérrez-Nava, Diana Reyes-Pavón, Alejandro Azaola-Espinosa, Lino Mayorga-Reyes

**Affiliations:** 1Doctorado en Ciencias Biologicas y de la Salud, Universidad Autonoma Metropolitana, Unidad Xochimilco, Ciudad de Mexico 04960, Mexico; 2202802526@alumnos.xoc.uam.mx; 2Laboratorio de Biotecnologia, Departamento de Sistemas Biologicos, CONAHCYT-Universidad Autónoma Metropolitana, Unidad Xochimilco, Ciudad de Mexico 04960, Mexico; mleyte@correo.xoc.uam.mx; 3Facultad de Medicina Mexicali, Universidad Autonoma de Baja California, Mexicali 21000, Mexico; edgar.torres.maravilla@uabc.edu.mx (E.T.-M.); diana.reyes.pavon@uabc.edu.mx (D.R.-P.); 4Laboratorio de Biotecnologia, Departamento de Sistemas Biologicos, Universidad Autonoma Metropolitana, Unidad Xochimilco, Ciudad de Mexico 04960, Mexico; 2182027221@alumnos.xoc.uam.mx (J.I.B.-H.); fmendoza@correo.xoc.uam.mx (F.M.-P.); azaola@correo.xoc.uam.mx (A.A.-E.); 5Centro de Investigación en Recursos Bioticos, Facultad de Ciencias, Universidad Autonoma del Estado de Mexico, Carretera Toluca-Ixtlahuaca Km 14.5, San Cayetano, Toluca 50295, Mexico; 6Unidad de Cuidados Intensivos de Neonatos, Unidad Medica de Alta Especialidad, Hospital Gineco Obstetricia No. 4 “Luis Castelazo Ayala”, Instituto Mexicano del Seguro Social, Ciudad de Mexico 01090, Mexico; leovigildo.mateos@imss.gob.mx; 7Laboratorio de Ecologia Microbiana, Departamento de Sistemas Biologicos, Universidad Autonoma Metropolitana, Unidad Xochimilco, Ciudad de Mexico 04960, Mexico; agutz@correo.xoc.uam.mx

**Keywords:** *Bifidobacterium longum*, antioxidant, BSH activity, genome, functional characterization

## Abstract

*Bifidobacterium longum* is considered a microorganism with probiotic potential, which has been extensively studied, but these probiotic effects are strain dependent. This work aims to characterize the probiotic potential, based on the biochemical and genomic functionality, of *B. longum* LBUX23, isolated from neonates’ feces. *B. longum* LBUX23 contains one circular genome of 2,287,838 bp with a G+C content of 60.05%, no plasmids, no CRISPR-Cas operon, possesses 56 tRNAs, 9 rRNAs, 1 tmRNA and 1776 coding sequences (CDSs). It has chromosomally encoded resistance genes to ampicillin and dicloxacillin, non-hemolytic activity, and moderate inhibition of *Escherichia coli* ATCC 25922 and to some emergent pathogen’s clinical strains. *B. longum* LBUX23 was able to utilize lactose, sucrose, fructooligosaccharides (FOS), and lactulose. The maximum peak of bacterial growth was observed in sucrose and FOS at 6 h; in lactose and lactulose, it was shown at 8 h. *B. longum* LBUX23 can survive in gastrointestinal conditions (pH 4 to 7). A decrease in survival (96.5 and 93.8%) was observed at pH 3 and 3.5 during 120 min. *argC*, *argH*, and *dapA* genes could be involved in this tolerance. *B. longum* LBUX23 can also survive under primary and secondary glyco- or tauro-conjugated bile salts, and a mixture of bile salts due to the high extracellular bile salt hydrolase (BSH) activity (67.3 %), in taurocholic acid followed by taurodeoxycholic acid (48.5%), glycocholic acid (47.1%), oxgall (44.3%), and glycodeoxycholic acid (29.7%) probably due to the presence of the *cbh* and *gnlE* genes which form an operon (start: 119573 and end: 123812). Low BSH activity was determined intracellularly (<7%), particularly in glycocholic acid; no intracellular activity was shown. *B. longum* LBUX23 showed antioxidant effects in DPPH radical, mainly in intact cells (27.4%). In the case of hydroxyl radical scavenging capacity, cell debris showed the highest reduction (72.5%). In the cell-free extract, superoxide anion radical scavenging capacity was higher (90.5%). The genome of *B. longum* LBUX23 contains *PNPOx*, *AhpC*, *Bcp*, *trxA,* and *trxB* genes, which could be involved in this activity. Regarding adherence, it showed adherence up to 5% to Caco-2 cells. *B. longum* LBUX23 showed in vitro potential probiotic properties, mainly in BSH activity and antioxidant capacity, which indicates that it could be a good candidate for antioxidant or anti-cholesterol tests using in vivo models.

## 1. Introduction

*Bifidobacterium longum* is a human symbiont bacterium considered one of the most common and prevalent species in the healthy human gut, whose richness changes according to age [1]. Three subspecies have been identified: *B. longum*, *Bifidobacterium infantis,* and *B. longum* subsp. *suis*, commonly found in the porcine gut, while *B. longum* subsp. *infantis* and *longum* have been isolated from infant and adult intestines, respectively [2,3]. Several probiotic properties have been described for *B. longum* strains in both in vitro experiments and in vivo animal studies. These have included antiviral, metabolic, antioxidant, immunomodulatory, and neuromodulating effects, among others. However, these effects are strain dependent [4].

The scientific evidence suggests that *B. longum* has antioxidant properties and can reduce oxidative stress in the intestine. Some bifidobacteria possess peroxide reductase (PrxR) and NADPH oxidases (NOXs), which are capable of efficiently eliminating reactive oxygen species (ROS) and enhancing the tolerance of bifidobacteria to oxygen. When bifidobacteria exhibit higher resistance to oxygen, their capacity for scavenging free radicals becomes more potent [5]. An excess of ROS interacts with several biomolecules, resulting in dysfunction and cellular damage, leading to structural and functional disorders, including cancer, neurodegenerative diseases, diabetes mellitus, atherosclerosis, heart and/or brain failure, rheumatoid arthritis, inflammatory bowel diseases, colon cancer, Alzheimer’s disease, and aging among others [6,7]. The antioxidant properties of probiotics include their ability to generate different antioxidant metabolites such as butyrate, glutathione (GSH), and folate. Probiotics can also hinder the growth of harmful intestinal pathogens and lower post-meal lipid levels, which contribute to oxidative damage. Since appropriate antioxidant mechanisms play a vital role in maintaining our body’s proper functioning, it is important to extensively study probiotics for their potential antioxidant benefits [7]. In this context, the plasma lipid profile is another important factor modulated by the probiotic strains through antioxidant and enzyme activities. According to Jiang et al. (2021), it was proposed that *B. longum* strains possess a unique capacity to individually influence cholesterol reduction. These effects are influenced by specific bacterial traits, such as their ability to deconjugate bile salts, assimilate cholesterol, and express genes involved in cholesterol metabolism [8]. The hypocholesterolemic activity of *B. longum* can be exerted through two different mechanisms: *i*) bile salt deconjugation by bile salt hydrolase and *ii*) cholesterol assimilation by specific absorption [9]. BSH is an enzyme responsible for breaking down glycine- and/or taurine-conjugated bile salts into amino acid residues and liberating bile acids through hydrolysis. Conjugated bile salts usually move through enterohepatic circulation; due to deconjugation by BSH, the bile salts become less soluble and are excreted by the feces, which generates a replacement of the excreted bile salts with new synthesized bile salts at the expense of stored cholesterol [10,11,12,13].

Among the criteria for selecting probiotics or candidate strains are the identification, safety, functionality characterization, and healthy benefits. Currently, the most suitable method for identifying probiotics and attributing their effects to specific microbial strains appears to be through the analysis of phenotypic and genotypic data. Therefore, properly identifying a probiotic strain requires a comprehensive assessment of its morphological, physiological, and biochemical characteristics, along with considerations of its genetic profile [14]. The functionality characterization involves assessing the ability to withstand various stress conditions, such as salivary/gastric enzymes, low pH, gastric juice, and bile. This evaluation helps to elucidate the potential health-promoting effects of the tested substances. The aim of this study was to characterize the probiotic potential, based on the biochemical and genomic functionality, of *B. longum* LBUX23, a strain isolated from neonates’ feces. To assign probiotic effects in vitro and to generate evidence so that in the future, it can be tested in an in vivo model and/or clinical trial and could be used in biotechnological applications, human health, and food science.

## 2. Materials and Methods

### 2.1. Isolation and Strain Propagation

This study was performed using stool samples from clinically healthy neonates delivered vaginally at Obstetrics Gynecology Hospital No. 4, "Luis Castelazo Ayala". The first sample was collected from birth to hospital discharge. This protocol was accepted by the Ethics Commission of the Mexican Social Security Institute (IMSS), document number 36068. Prior to inclusion, written informed consent was obtained from each donor sample’s parent or guardian, ensuring that fecal samples were collected under ethical conditions during that time. The samples (1 g) were diluted in 1 mL of reduced-PBS (phosphate-buffered saline) and kept at −80 °C until analyses were conducted. Subsequently, the samples were plated in triplicate onto TPY medium agar plates, which were incubated anaerobically in an anaerobic chamber (85% nitrogen, 10% hydrogen, 5% carbon dioxide) (Forma anaerobic system Model 1025, Thermo Scientific, Waltham, MA, USA) at 37 °C for 48 h. The chosen isolates were subjected to optical microscopy for the examination of their morphology and Gram staining outcomes. Furthermore, their catalase activity was assessed. The Gram-positive and catalase-negative isolates with typical bifidobacteria shapes were genotypically identified [15]. The propagation of the *Bifidobacterium* strains was performed in TPY medium in an anaerobic chamber at 37 °C for 24–48 h.

### 2.2. DNA Extraction and Genotypic Identification

The Wizard^®^ Genomic DNA Purification Kit (Promega, Tokyo, Japan) was used to isolate total genomic DNA according to the manufacturer’s instructions. The 16S rRNA sequencing was conducted at the Divisional Molecular Biology Laboratory of Universidad Autonoma Metropolitana campus Iztapalapa, Mexico City, using the Bif 164 (GGGTGGTAATGCCGGATG) and Bif 662 (CCAC-CGTTACACCGGGAA) primers [16]. Comparisons and sequence alignments were performed using MEGA5 and NCBI’s basic local alignment search tool (http://blast.ncbi.nlm.nih.gov/Blast.cgi (accessed on 16 January 2023)).

### 2.3. Multilocus Sequence Analysis (MLSA)

A comparison of constitutive genes in different strains of *B. longum* strains was made: *tlyC*, *cbh*, *glnE*, *pNPOx*, *ahpC*, *trxA*, *trxB* [11,17,18]. The MLSA approach has been used to generate a robust and highly discriminatory supertree to infer phylogeny in the genus *Bifidobacterium*. MLSA was performed according to González–Vázquez et al., 2022 [18].

### 2.4. Genomic analysis and annotation

The sequencing was conducted at the Integrated Microbiome Resource (IMR) located t Dalhousie University in Nova Scotia, Halifax, NS, Canada [17]. Assembling was carried out by IMR using SMRT analysis software v12.0. Data processing, plot construction, and genome annotation were performed according to González–Vázquez et al., 2022 [18].

### 2.5. Effect of Carbon Source on the Growth of B. longum LBUX23

The growth kinetics were monitored for a duration of 8 h of fermentation, representing the time required for the bacteria to reach the stationary phase. TPY medium was utilized, and various substances, including lactose, sucrose, fructooligosaccharides, and lactulose, were independently added at a concentration of 1% *w*/*v*. All media were adjusted to a pH of 7 using 0.1 M NaOH (Thermo Scientific Orion 410A+, USA). Prior to sterilization, the medium was purged of oxygen by bubbling CO_2_. Samples were collected at regular intervals of 2 h from time zero to 8 h. The bacterial concentration was determined by the plate count method [19]. The pH of the supernatant was measured according to González–Vázquez et al., 2022 [18]. All experiments were completed in triplicate.

### 2.6. Antibiotic Profile

Multidisc PT-34 Multibac I.D. (Investigación Diagnostica, Ciudad de Mexico, Mexico) were used to determine the ability to resist antibiotics following the supplier’s instructions. The antibiotics used in the study were as follows: β-lactam antibiotics, including vancomycin (30 μg), ampicillin (10 μg), dicloxacillin (1 μg), cephalothin (30 μg), penicillin (10 U), and cefotaxime (30 μg); inhibitors of protein synthesis such as gentamicin (10 μg), clindamycin (30 μg), erythromycin (15 μg), and tetracycline (30 μg); inhibitors of nucleic acid synthesis such as ciprofloxacin (5 μg) and other compounds such as trimethoprim-sulfamethoxazole (25 μg). The antibiotic profile of *Bifidobacterium animalis* subsp. *lactis* Bb-12 (CHR Hansen, Hørsholm, Denmark) was also determined for comparison. All experiments were conducted in triplicate [18].

### 2.7. Hemolysis Test

Prior to the experiment, *B. longum* LBUX23 was cultured overnight in MRS broth supplemented with 0.5% (*w*/*v*) L-cysteine. A concentration of 1 × 10^9^ CFU/mL of the bacterial sample was inoculated onto blood agar plates and then incubated for 48 h at 37 °C in an anaerobic chamber. The test was performed in triplicate for each strain. *Staphylococcus aureus* ATCC 6538 served as the positive control, while *Bifidobacterium animalis* subsp. *lactis* Bb-12 (CHR Hansen) was used as the negative control. Results were reported as follows: positive indicating total hemolysis, and negative as non-hemolytic.

### 2.8. Antimicrobial Activity

The strain’s capacity to inhibit various bacterial strains, including *Escherichia coli* ATCC 25922, *Escherichia coli* O157:H7, *Salmonella typhi* ATCC 14028, *Staphylococcus aureus* ATCC 6538, as well as emergent pathogens obtained from clinical samples such as *Klebsiella pneumoniae*, *Proteus mirabilis*, and *Pseudomonas aeruginosa*, was assessed using the agar well diffusion assay. According to antibiotic susceptibility, these strains were classified as multi-drug resistant (MDR) and extremely resistant (XDR), following the general criteria of Magiorakos et al. (2012) [20]. XDR strains were categorized as strains that exhibited resistance to at least one agent in all or two therapeutic election categories (referred to as groups A and B) [20]. *B. longum* LBUX23 and *B. animalis* Bb-12 (control) were cultivated on TPY agar at a concentration of 1 × 10^7^ CFU/mL and incubated at 37 °C for 24 h. Following this, warm, soft tryptic soy agar (8 g/L) was poured over each *Bifidobacterium* culture. Once solidified, other bacteria were individually plated onto the agar at a concentration of 1 × 10^8^ CFU/mL. The plates were then incubated in anaerobic conditions at 37 °C for 24 h. The results were interpreted as follows: absence of inhibition (−), weak inhibition (+), and strong inhibition (++).

### 2.9. Bile Salt Tolerance Assay

Fresh bacterial cultures were inoculated onto TPY agar supplemented with different concentrations (0.1%, 0.2%, 0.3%, and 0.5% *w*/*v*) of glycocholic acid, glycodeoxycholic acid, taurocholic acid, taurodeoxycholic acid, or oxgall (obtained from Sigma Aldrich, St. Louis, MO, USA). The agar plates were then incubated anaerobically at 37 °C for 48 h. Strains exhibiting bile salt hydrolase activity were characterized by the presence of a halo surrounding them, indicating the precipitation of deconjugated bile salts [21] and demonstrating their resistance. *B. animalis* subsp. *lactis* Bb-12 was employed as the positive control in this experiment.

### 2.10. Bile Salt Hydrolase Activity

This activity was determined in intracellular cell-free extracts and cell debris. The method described by González–Vázquez et al. (2015) [22] was used to quantify BSH activity. First, *B. longum* LBUX23 was inoculated at a concentration of 1 × 10^9^ CFU/mL in sodium phosphate buffer (PBS) supplemented with 0.5% *w*/*v* of glycocholic acid, glycodeoxycholic acid, taurocholic acid, taurodeoxycholic acid oxgall or any bile salt as control and incubated for 48 h. Following incubation, a 1 mL portion of the culture was subjected to centrifugation at 29,286× *g* for 5 min at 4 °C. Subsequently, 50 μL of the resulting supernatant was combined with 50 μL of sodium PBS (0.1 M, pH 6.0), 100 μL of a solution containing each bile salt or glycine (used as a control group) at a concentration of 0.5% *w*/*v*, and 10 μL of dithiothreitol (DTT). The mixture was incubated at 37 °C for 30 min to stop the enzyme activity, and 100 μL of trichloroacetic acid (15% *w*/*v*) was added. Afterward, the mixture was subjected to centrifugation for 5 min at 29,286× *g* and 4 °C. Then, 50 μL of the resulting supernatant was combined with 50 μL of distilled water and 1.9 mL of ninhydrin reagent. The ninhydrin reagent consisted of 0.5 mL of 1% *w*/*v* ninhydrin in citrate buffer (0.5 M, pH 5.5), 1.2 mL of glycerol, and 0.2 mL of buffer solution (0.5 M citrate, pH 5.5). The mixture was boiled for 14 min, followed by cooling for 3 min in an ice bath. The absorbance at 570 nm was measured. Prior to each assay, a standard curve was prepared using glycine and 10 μL of DTT to determine the concentration of released glycine. The percentage of BSH activity was calculated with the following:$$\mathrm{BSHactivity}\left(\%\right)=\left(\frac{{\mathrm{Concentration}}_{\mathrm{control}\;\mathrm{group}}}{{\mathrm{Concentration}}_{\mathrm{sample}}}\right)\times 100$$

### 2.11. pH Tolerance Assay

*B. longum* LBUX23 was cultured in TPY medium and incubated at 37 °C for 48 h. The culture was centrifuged at 10,000× *g* for 5 min, and the cells were washed twice using PBS buffer. The cells (1 × 10^9^ CFU/mL) were inoculated into TPY broth adjusted to different pH 1, 1.5, 2, 2.5, 3, 3.5, 4, 4.5, 5, 5.5, 6, 6.5, and 7 (control) with HCL 1N or NaOH 1N, respectively (Thermo Scientific Orion 410A+, USA). The media was incubated at 37 °C for 120 min and the bacterial concentration was determined by the plate count method [19]. The growth in CFU/mL was determined at 5, 10, 15, 20, 30, 60, and 120 min, respectively. All experiments were completed in triplicate. The survival rate of the strains at different pH and times was calculated as described above using the following expression according to Buntin et al. (2008) [23]:$$\mathrm{pH}\;\mathrm{survival}\;\mathrm{rate}\left(\%\right)=\left(\frac{\frac{\mathrm{CFU}}{\mathrm{mL}}120\;\min}{\frac{\mathrm{CFU}}{\mathrm{mL}}\;0\;\min}\right)\times 100$$

### 2.12. DPPH Radical Scavenging Activity Assay

The scavenging activity of the 2,2-diphenyl-1 picrylhydrazyl (DPPH) radical was assessed utilizing the procedure outlined by Su et al. (2015) [24]. In summary, a mixture was prepared by combining 1 mL of freshly prepared DPPH solution (0.2 mmol/L in ethanol) with 1 mL of the sample, which included intact cells, intracellular cell-free extracts, or cell debris. Subsequently, the mixture was incubated in darkness for 30 min. The absorbance of the solution was then determined at a wavelength of 517 nm. The DPPH scavenging ability was calculated as follows:$$\mathrm{Scavenging}\;\mathrm{activity}\;\left(\%\right):\left(1-\frac{A_{\mathrm{sample}}-A_{\mathrm{blank}}}{A_{\mathrm{control}}}\right)\times 100$$

### 2.13. Hydroxyl Radical Scavenging Activity Assay

The hydroxyl radical scavenging ability was analyzed according to Yan et al. (2020) [25]. Intact cells, intracellular cell-free extracts, or cell debris (1 mL per sample) were mixed with 1.0 mL of 1,10-phenanthroline solution (2.5 mM), 1 mL of PBS (pH 7.4), and 1 mL of FeSO_4_ (2.5 mM). To inhibit the reaction, 1 mL of 20 mM H_2_O_2_ was added, and the reaction was incubated at 37 °C for 90 min. The absorbance of the solution was measured at 517 nm. The scavenging ability of hydroxyl radicals was determined as follows:$$\mathrm{Scavenging}\;\mathrm{activity}\;\left(\%\right)=\frac{\left(A_{\mathrm{sample}}-A_{\mathrm{blank}}\right)}{\left(A_{\mathrm{control}}-A_{\mathrm{blank}}\right)}\times 100$$

### 2.14. Superoxide Anion Radical Scavenging Activity Assay

The superoxide anion radical scavenging ability was evaluated following the method of Yan et al. (2020) [25], with some modifications. Intact cells, intracellular cell-free extracts, or cell debris (1mL per sample) were added to 3 mL of Tris–HCl solution (pH 8.2) and incubated at 25 °C for 20 min. Then, 0.4 mL of pyrogallol (25 mM) was added at room temperature for 4 min. The reactions were stopped by adding 0.5 mL of HCl, and the absorbance was measured at 325 nm. The superoxide anion radical scavenging activity was determined as follows:$$\mathrm{Scavenging}\;\mathrm{activity}\;\left(\%\right)=\left(1-\frac{A_{\mathrm{sample}}}{A_{\mathrm{blank}}}\right)\times 100$$

### 2.15. Adhesion Assay

Caco-2 cells (ATCC^®^ HTB-37™, Sigma Aldrich, USA) were used to determine the cell adhesion ability of *B. longum* LBUX23 by utilizing the methodology reported in Gonzalez-Vazquez et al. (2022) [18]. Caco-2 cells were cultured in Eagle’s minimal essential medium (obtained from Gibco Invitrogen, Carlsbad, CA, USA) supplemented with 20% fetal bovine serum (FBS; Gibco Invitrogen, Carlsbad, CA, USA), 1% penicillin/streptomycin, 1% L-glutamine, and 1% (*v*/*v*) nonessential amino acids solution (Gibco). The cells, with a density of 1 × 10^5^, were seeded into 24-well culture plates and incubated at 37 °C with 5% CO_2_. After 24 h of incubation, a monolayer of cells was formed, and the plates were further incubated for a period of 15 days. To prepare the bacterial samples, an overnight culture of either *B. longum* LBUX23 or *B. animalis* subsp. *lactis* Bb-12 (utilized as a control) was subjected to centrifugation at 6000× *g* for 5 min at 4 °C. The resulting pellet was then washed twice with PBS (pH 7.4) and suspended in Dulbecco’s Modified Eagle Medium without antibiotics and fetal bovine serum. In each well of the 24-well plates, a bacterial suspension of 1 mL containing 1 × 10^8^ CFU/mL was introduced and incubated for 1 h at 37 °C in a 5% CO_2_ environment. Subsequently, each well was subjected to three washes with PBS (pH 7.4) to eliminate non-adherent bacteria. The bacteria that adhered to the monolayer were detached using a solution of trypsin-EDTA (obtained from Gibco, Carlsbad, California, USA) in PBS. The enzymatic activity was deactivated by adding a culture medium. The bacterial samples were subjected to serial dilutions, and the resulting dilutions were plated onto MRS agar plates containing 0.5% (*w*/*v*) L-cysteine hydrochloride (obtained from Merck, Rahway, NJ, USA). After 48 h of incubation in anaerobic conditions, the bacterial colonies were counted. All experiments were performed in triplicate.

### 2.16. Statistical Analysis

Significant differences in growth by using different carbon sources and bile salt in regard to DPPH scavenging ability, hydroxyl scavenging ability, and anion superoxide scavenging were determined by ANOVA (*p* ≤ 0.05). In the case of adhesion assay, differences were determined by *t*-student (*p* ≤ 0.05) using GraphPad Prism software version 5.01 (GraphPad Software, Inc., San Diego, CA, USA).

## 3. Results

### 3.1. Isolation and Genotypic Identification

In this study, we isolated one-Gram-positive rod-shaped bacterium; according to the BLAST analysis, those strains showed 99% of identity with the *Bifidobacterium* genus. The multilocus analysis showed that it belongs to *longum* subspecies (Figure 1) and was named *B. longum* LBUX23.

The complete genome was registered at GenBank CP116390 as *B. longum* LBUX23. The BioProject accession number is PRJNA924960.

### 3.2. Genome Analysis

The genomic analysis showed that *B. longum* LBUX23 has a plasmid-free genome with a single 2,287,838 bp circular chromosome with a G+C content of 60.1%. The genomic annotations illustrated a total of 1853 coding sequences. The genome of *B. longum* LBUX23 possesses 56 tRNAs, 9 rRNAs, 1 tmRNA, and 1776 CDSs (95.84%). These last were grouped according to their functionality in cellular processes and signaling: (17.54%), information storage and processing (23.98%), metabolism (36.19%), and poorly characterized (22.29%). After COG categories were assigned, 22.29% were genes of unknown function, 9.51 % belonged to the carbohydrate transport and metabolism categories, 9.14% to amino acid transport and metabolism, 8.56 % to transcription, 7.82% to replication, recombination, and repair, 7.55% to translation, ribosomal structure, and biogenesis, 4.38% inorganic ion transport and metabolism, 4.23% nucleotide transport and metabolism, 3.96% cell wall, membrane, envelope biogenesis, 3.38% coenzyme transport and metabolism, 3.28% signal transduction mechanisms, 3.12% defense mechanisms, 2.96% energy production and conversion, 2.54% intracellular trafficking, secretion, and vesicular transport, 2.27% posttranslational modifications, protein turnover and chaperones, 2.01% lipid transport and metabolism, 1.95% cell cycle control, cell division, and chromosome partitioning, 0.58% secondary metabolites biosynthesis, transport, and catabolism, 0.42% cell motility and 0.05% chromatin structure and dynamics. *B. longum* LBUX23 did not contain clustered regularly interspaced short palindromic repeats (CRISPR).

### 3.3. Safety and Functional Characterization of B. longum LBUX23

#### 3.3.1. Effect of Carbon Source on Growth

The *B. longum* LBUX23 grew in all carbon sources (Figure 2), initiating the exponential phase between 4-8 h. The maximum peak of bacterial growth was observed in sucrose and FOS at 6 h and at 8 h in lactose and lactulose **(**Figure 2). The pH decreased to 1.27 in the sucrose medium, followed by FOS (0.86), lactulose (0.65), and lactose (0.49), with respect to the initial pH of each substrate (Figure 2). However, no significant differences were found between them.

#### 3.3.2. Antibiotic Profile

In terms of the antibiotic profile, both *B. longum* LBUX23 and *B. animalis* Bb-12 exhibited resistance to ampicillin and dicloxacillin. However, *B. animalis* Bb12 also showed resistance to cephalothin tetracycline and ciprofloxacin (Table 1a).

#### 3.3.3. Hemolysis Test

Table 1b shows the results of the hemolysis activity. *B. longum* LBUX23 and *B. animalis* Bb-12 demonstrated negative hemolysis activity.

#### 3.3.4. Antimicrobial Activity

Table 1c presents the results of the antimicrobial activity testing of *B. longum* LBUX23 and *B. animalis* Bb-12 against various strains.

#### 3.3.5. Bile Salt Tolerance Assay and Activity

When *B. longum* LBUX23 and *B. animalis* Bb-12 grew in different concentrations and types of bile salts (Table 1d), a halo of precipitation was observed. *B. longum* LBUX23 showed significant differences in extracellular bile salt hydrolase activity between taurocholic acid (67.3%) and glycocholic acid (47.8%), oxgall (44.3%), and glycodeoxycholic acid (29.7%) (*p =* 0.0494, *p =* 0.0247, and *p <* 0.001, respectively) (Figure 3a). In addition, a higher percentage of extracellular BSH activity in taurocholic acid to *B. longum* LBUX23 (67.3%) in comparison to *B. animalis* Bb-12 (13.9%) (*p <* 0.0001) was determined (Figure 3a). In the case of intracellular BSH activity, no activity in glycocholic acid was shown by *B. longum* LBUX23 and *B. animalis* Bb-12 (Figure 3a). *B. longum* LBUX23 showed the highest BSH activity in taurodeoxycholic acid (8.2%), followed by taurocholic acid (6.3%) in comparison to *B. animalis* Bb-12, which showed the highest BSH activity in glycodeoxycholic acid (4.2%) acid and oxgall (2.9%). No significant difference in intracellular activity between *B. longum* LBUX23 and *B. animalis* Bb-12 was shown (Figure 3b). The genomic analysis showed the presence of the *cbh* gene, which codify to BSH (EC 3.5.1.24), as a part of an operon containing two genes, *cbh,* and *glnE*, which in turn codify to a glutamine synthetase adenylyltransferase (E.C. 2.7.7.42 2.7.7.89); this operon might be responsible for BSH activity.

#### 3.3.6. pH Tolerance Assay

The survival rate of *B. longum* LBUX23 at different pH is shown in Figure 4a. *B. longum* LBUX23, did not survive to pH 1-2.5. In the case of pH 3 and 3.5, a decrease in survival rates to 96.5 and 93.8% were observed. An increase of 20.1%, 22.8%, 30.8%, 50.8%, 77.8%, 89.6%, and 117.8% in the percentage of survival with respect to pH 3.5 was shown to pH 4, 4.5, 5, 5.5, 6, 6.5, and 7, respectively (Figure 4a). When survival was tested at pH 1 after 5 min, a 99.3% decrease was observed for *B. longum* LBUX23, while at pH 1.5, the same percentage of reduction was observed but after 10 min. In the case of pH 2, a decrease in survival of 98.8% was shown after 30 min. At pH 2.5, we observed a decrease of 99% after 120 min of submission (Figure 4b). The genomic analysis of *B. longum* LBUX23 showed the presence of *argC* (NCBI Locus tag: PIB40_02085), *argH* (PIB40_02045), and *dapA* (PIB40_02760) genes, which have been reported as responsible for the ability to tolerate acid environments in other species of *B. longum* [26].

#### 3.3.7. Antioxidant Activity

*B. longum* LBUX23 showed more DPPH radical scavenging activity than *B. animalis* Bb-12 in cell-free extracts (5.2 and 2.3%, respectively, *p* < 0.0095) and intact cells (27.4 and 24.2%, respectively, *p* < 0.0038). In contrast, cell debris of *B. animalis* Bb-12 showed higher antioxidant ability (17.9%) than *B. longum* LBUX23 (13.5%) *p* < 0.0003 (Figure 5a). By searching in the genome deposited at GenBank, we found *nrdH* (PIB40_05210), *msrAB* (PIB40_04685), *PNPOx* (PIB40_09440), *AhpC* (PIB40_00775), *Bcp* (PIB40_00775), trxA (PIB40_02940), and trxB (PIB40_04850) genes, which have been previously reported as possibly responsible of antioxidant activity involved in antioxidant activity (Table 2) [5,27,28].

The ability to eliminate the hydroxyl radical was determined for *B. longum* LBUX23 and *B. animalis* Bb-12 (Figure 5b). We observed that intact cells of *B. longum* LBUX23 showed more hydroxyl activity (19.9%) than *B. animalis* Bb-12 (15.6%). Regarding cell-free extract and cell debris, hydroxyl radical scavenging activity was higher in *B. animalis* Bb-12 (45.6 and 76.3%, respectively) than in *B. longum* LBUX23 (40.2 and 72.5%, respectively). However, no significant differences among strains (Figure 5b) were shown. Cell debris showed major hydroxyl activity followed by cell-free extract and intact cell (*p* < 0.0001) in *B. longum* LBUX23 and *B. animalis* Bb-12 (Figure 5b). In the genome, we found some genes related to this activity, which are described in Table 2.

In this study, we showed that *B. longum* LBUX23 and *B. animalis* Bb-12 could inhibit superoxide anion radicals (Figure 5c). *B*. *longum* LBUX23 showed more activity in cell debris (25.3%) and intact cells (78.6%) in comparison to *B. animalis* Bb-12 (23.2 and 75.6%, respectively). In the case of cell-free extract, higher activity was found in *B. animalis* Bb-12 (92.8%) than in *B. longum* LBUX23 (90.5%). However, no significant differences were found for both strains in the case of cell-free extract, cell debris, and intact cells of both strains (Figure 5c). In addition, we observed that *B. longum* LBUX23 and *B. animalis* Bb-12 had major activity in cell-free extracts, followed by activity in cells and cell debris. Significant differences were found to cell debris in comparison to cell-free extracts (*p <* 0.0001) and intact cells (*p <* 0.0001) in both strains (Figure 5c). Genome analysis showed the presence of some genes involved in antioxidant activity (Table 2).

#### 3.3.8. Adhesion Assay

It was observed that *B. longum* LBUX23 could adhere up to 5% to the human epithelial cells Caco-2, whereas *B. animalis* subsp. *lactis* Bb-12 was able to adhere up to 7.12%. However, neither strain showed significant differences (*p >* 0.05) (Figure 5d). In addition, genomic annotation of *B. longum* LBUX23 strain showed the presence of *lspA* (PIB40_08330), *dnaK* (PIB40_04505), *grpE* (PIB40_00120), *aprE* (PIB40_04245), *dppB3* (PIB40_02590), sortase family (PIB40_04400 and PIB40_04760) and fibronectin type III-like domain (PIB40_05180 and PIB40_07405) genes, might be responsible for adhesion as have been previously reported [29].

## 4. Discussion

The population of *Bifidobacterium* is a common inhabitant of healthy humans. The alteration in the number and composition of *Bifidobacterium* population present in the human microbiome has been associated with several diseases, such as irritable bowel syndrome, inflammatory bowel disease (IBD), obesity, and allergy, among others. The consumption of bifidobacteria in various clinical trials has demonstrated promising outcomes in several clinical aspects, including the prevention of diarrhea, alleviation of symptoms in ulcerative colitis and irritable bowel syndrome (IBS), and the prevention of necrotizing enterocolitis [30,31]. *Bifidobacterium* genus is present in early life and plays an important role in newborns and infant development, which will determine an individual’s health at later stages of life [32]; *Bifidobacterium* species are most abundant in infants from the first week to the sixth month of life. However, it should be noted that not all fecal samples contain isolates of *Bifidobacterium* strains [33]. In this study, we found one *Bifidobacterium* bacteria (Figure 1) belonging to *longum* species from a fecal sample of a newborn. The genomic size and G+C content were similar to the one reported to *B. longum* LTBL16 [5], *B. longum* subsp. *longum* 35624 [34], *B. longum* GT15 [35], *B. longum* subsp. *longum* D4, M12, E1, S3, [3] and another strain reported by Arboleya et al. (2018) [36]. Research on bifidobacteria has utilized various bioinformatic tools to examine their complete genomes, providing valuable insights into the mechanisms through which these bacteria adapt to the unique conditions of the gastrointestinal tract. These genome characterizations have also revealed probiotic functions that facilitate specific interactions between the host and microorganisms [37]. Therefore, the genomic characterization of bacteria with probiotic potential is necessary and important, including newly isolated probiotics [38]. According to the genomic analysis, we showed that no CRISPR system in the *B. longum* LBUX23 was found, in comparison with other strains of *B. longum* (9, 17-1B, 105-A, BG7, and GT15) [39]. In 2015, Briner et al. (2015) [40] reported several strains of the *Bifidobacterium* that contained CRISPR sequences and concluded that the presence of CRISPR systems might be strain- rather than species-dependent [40]. The CRISPR system represents the strain’s record of immunity and the environmental challenges suffered by invasive DNA [31]. In the case of *Bifidobacterium* strains, the CRISPR targeting prophages are present in the genome of several bifidobacterial species (bifidophages). These findings indicate that these species reside in the same ecological habitat, where coevolution between CRISPR immune systems and prophages has likely taken place [31]. We hypothesize that *B. longum* LBUX23 has not been exposed to invasive DNA or infection and interaction with bifidophages. In addition, it has been suggested that the occurrence of CRISPR systems in the *Bifidobacterium* genus was 77.2% [40]. In the case of *B. longum* subsp. *longum* strains, the occurrence of CRISPR has been around 44.2% [41]. Bifidobacteria usually have diverse and extensive CRISPR systems. A relationship was noted between strains without CRISPR systems and the frequency of targeting of prophages in their chromosome by other CRISPR spacers [40]. 

The *Bifidobacterium* genus can metabolize a wide variety of mono-, di-, and oligosaccharides that are present in the intestinal environment. These carbohydrates are imported into their cytoplasm through ABC transporters [42]. In this study, we found that 9.51% of COG, are involved in metabolizing and transporting carbohydrates. In addition to *B. longum,* LBUX23 could be grown on different carbon sources, including glucose (data not shown), sucrose, lactose, and lactulose (Figure 2), and likely, this ability to metabolize carbohydrates is via bifid-shunt, as has been reported in *B. longum* NCC2705 [18,43]. Additionally, *B. longum* LBUX23 was grown in fructooligosaccharides (FOS), which are non-digestible carbohydrates, that can be found in various plant-based sources such as onions, asparagus, artichokes, garlic, wheat, bananas, tomatoes, and honey [44]. FOS are one of the significant classes of bifidogenic oligosaccharides and is one of the established prebiotics, defined as "a substrate that is selectively utilized by *Bifidobacterium* conferring a health benefit" [45,46]. Additionally, in the genomic analysis, we found some genes involved in carbohydrates metabolism, such as *GalA1* (PIB40_05300), *Ldh* (PIB40_00085 and PIB40_08485), *Ppc* (PIB40_04825), *GlGP* (PIB40_04785), *Fba* (PIB40_04610), *TreY* (PIB40_04580), *AmyA* (PIB40_01230), *MalQ1* (PIB40_04535), *LeuA* (PIB40_04350), and several genes to code by glycosyl hydrolase. These results showed a great diversity of genes and glycosyl-hydrolases, which confer the ability to metabolize a wide range of sugars [18,36,42]. These results were similar to those reported by González–Vázquez et al. (2022) [18], who grew *B. pseudocatenulatum* JCLA3 on glucose, lactose, sucrose, or inulin and found 44 predicted glycosyl hydrolases genes, which can act over carbohydrates metabolism [18]. In addition, Blanco et al. (2020) [47] found the presence of genes coding glycosyl hydrolases in *B. longum* subsp. *longum* [47].

The European Food Safety Authority (EFSA) suggests that assessing hemolytic activity is important when choosing probiotic strains. These strains are considered non-virulent, making them suitable for use in food products [48]. In this study, the hemolytic activity of *B. longum* LBUX23 was evaluated. Neither α-hemolytic nor β-hemolytic activity was shown in blood agar plates (Table 1). Our results agreed with the findings of Kim et al. (2018) [49] as they did not show hemolysis in *B. bifidum* BGN4 and *B. longum* BORI [49]; and Yasmin et al. (2020) [48], who also evaluated hemolytic activity as sensitive to antibiotics [48]. The *B. longum* LBUX23 has been resistant to β-lactam antibiotics, like other *Bifidobacterium* species such as *B. pseudocatenulatum* B700 and *B. pseudocatenulatum* JCLA3 [18,50]. However, in the genome of *B. longum* LBUX23, we found that resistance to antibiotics is chromosomal since the genome did not present plasmids, then β-lactam resistance cannot be transferred by conjugation to other bacteria. However, the possibility exists that *B. longum* LBUX23 and other generally recognized as safety (GRAS) bacteria could transfer the genomic material by transformation mechanism; therefore, it is necessary to define the GRAS status again from a molecular point of view.

In addition, *oppA* (PIB40_00680 and PIB40_008910), *oppB* (PIB40_08915), *oppC* (PIB40_08920), and *oppF* (PIB40_08925) are found in its genome, which are involved in Quorum sensing (Qs) and resistance to β-lactam, as reported by González–Vázquez et al. (2022) [18], who found intrinsic resistance to β-lactam genes in *B. pseudocatenulatum* JCLA3 [18,50]. 

In the current study, the antimicrobial activity of *B. longum* LBUX23 was investigated against ATCC strains and clinical isolates (Table 1). These last were categorized according to their antibiotic resistance pattern [20]. We found weak inhibition of *K. pneumoniae* MDR, *P. mirabilis* MDR, *P. aeruginosa* XDR, and *E. coli* ATCC 25922. The antimicrobial effects observed are likely attributed to the synthesis of antimicrobial compounds, including organic acids, hydrogen peroxide, ethanol, diacetyl, acetaldehyde, saturated or unsaturated free fatty acids, and other substances such as peptides and bacteriocins [51,52,53,54]. Additionally, we hypothesized that the possible advantage of this strain is the activity against emerging pathogens shown, not in the regard of being used during the infection, but instead in the sense of being used as a probiotic favoring self-defense from the gastrointestinal tract, decreasing the probability of an emerging pathogen attacking the host; however, this would have to be demonstrated in an *in vivo* study. Furthermore, a metabolomic study is required when the *B. longum* LBUX23 grows in a coculture with the pathogen to find postbiotics. Postbiotics have been demonstrated to possess various biological activities, including antimicrobial, antioxidant, anti-inflammatory, antiproliferative, and immunomodulatory effects [55]. 

The potential of the isolate as a probiotic was evaluated by assessing its ability to survive under simulated conditions mimicking the digestive tract [48,56]. Strain selection typically relies on assessing the tolerance of probiotics to physiologically relevant stresses, including low pH and bile [57]. The pH level is a critical factor that affects the growth and viability of probiotics during their journey through the stomach [58]. Bifidobacteria, being moderately acid-tolerant microorganisms, exhibit optimal growth and viability at pH levels ranging from 6.5 to 7.0. However, these commensal microbes can grow at a pH higher than 4.5 [59] since they are natural inhabitants of the large intestine, where the pH begins to be more basic with respect to the stomach and small intestine. We showed that *B. longum* LBUX23 survived optimally in a range of pH 4 to 7 (Figure 4). Therefore, in the case of using this bacterium as a supplement, it should be administered after consuming food to previously increase gastric pH or by using a matrix that protects it from stomach acid to ensure its viability until it reaches the large intestine [60,61]. In the case of pH 3 and 3.5, *B. longum* LBUX23 is capable of surviving < 10%, probably due to the presence of *argC* (PIB40_02085), *argH* (PIB40_02045), and *dapA* (PIB40_02760) genes. These genes have been previously reported by Sundararaman et al. 2021 [26] as those that confer to *B. longum* NCIM 5672 a survival advantage in an acid environment (pH 3). In the case of *B. Longum* LBUX23, those genes could be responsible for acid tolerance; nevertheless, more research is needed to prove their expression and to elucidate how the proteins codified by those genes act against acidity.

The ability of probiotics to tolerate bile salts has often been included among the criteria for probiotic strain selection [57]. Bile salt hydrolase catalyzes the hydrolysis of the amide bond in conjugated bile salts by choloylglycine hydrolase (EC 3.5.1.24.), resulting in the release of free amino acids, which makes the bile salt insoluble and is finally secreted through stools. Bile salts are primarily produced by combining cholesterol with amino acids glycine or taurine in the liver. These synthesized bile salts are then stored in the gallbladder until they are released into the duodenum following the consumption of fatty foods [62]. The identification of BSH activity has been included as a criterion for probiotic strain selection. Currently, there is a growing focus on investigating the bacterial conversion of bile in the human gastrointestinal tract due to its potential role in the development or prevention of metabolic and inflammatory conditions [63]. Our study shows that *B. longum* LBUX23 has the presence of 2.01 % of COG assigned to lipid transport of metabolism and includes the choloylglycine hydrolase gene. This gene can metabolize primary and secondary biliary salts (Figure 3), such as another *Bifidobacterium* species [10,18,64,65]. In addition, Begley et al. (2006) [57] suggest that the ratio of glycoconjugated to tauroconjugated bile salts is 3:1 in enterohepatic circulation [57]. In our study, *B. longum* LBUX23 had high extracellular BSH activity (67.3%) in taurocholic acid, followed by taurodeoxycholic acid (48.5 %), glycocholic acid (47.1%), oxgall (44.3 %), and glycodeoxycholic acid (29.7 %). Likely, the high activity in a taurocholic acid or taurodeoxycholic acid was due to *tlyC1* ( PIB40_08320) gene present in *B. longum* LBUX23. The gene in question encodes a protein like hemolysin, which enhances the resistance of *B. longum* BBMN68 to bile acids conjugated with taurine, as documented in previous studies [66]. The intracellular activity was not found for glycocholic acid and less than 7 % for the other bile salts. These results suggest that BSH activity is extracellular, such as the results shown by Morinaga et al. (2022) [67,68]. Due to the high concentration of bile salts present in the intestine, some bacteria synthesize the BSH enzyme, and the BSH activity can decrease the toxicity of conjugated bile acids for them. The deconjugation of bile acids results in a reduction in their solubility and detergent properties, which can potentially decrease their toxicity to intestinal bacteria [68,69]. Bifidobacteria was able to produce precipitates in agar plates supplemented with biliary salt; due to the deconjugation of biliary salts by the BSH enzyme [67]. This effect is associated with probiotic cholesterol-lowering properties [70]. Another reason why *B. longum* LBUX23 has high BSH activity could be due to a previous adaptation since the source of its isolation was the newborn’s stools. Jarocki et al. (2014) [65] and Tanaka et al. (1999) [71] independently suggested that the activity of BSH is closely associated with the natural habitat of bacteria. Strains that exhibit BSH activity are typically found in the intestinal environment, where they are exposed to bile salts [65,71]. Additionally, Bordoni et al. (2013) [72] conducted a study evaluating cholesterol assimilation and BSH activity in 34 *Bifidobacterium* strains of human origin. They observed that strains belonging to *B. animalis*, *B. breve*, *B. longum*, and *B. pseudocatenulatum* species showed higher levels of BSH activity. Notably, *B. longum* subsp. *longum* MB 214 demonstrated the highest activity, followed by *B. breve* MB 11. These results are like the activity shown by *B. longum* LBUX23; however, it is necessary to homogenize the determination of BSH activity among the different probiotic strains in order to really be able to compare among them. Finally, high levels of activity of BSH are interesting as correlated with the ability to lower serum cholesterol levels in hypercholesterolemic [12]. Therefore, *B. longum* LBUX23 may be a candidate strain to test this effect in an in vivo model.

The presence of an excess of ROS significantly contributes to cell damage, including cell membrane injury, protein denaturation, and erroneous DNA replications. These effects can induce aging and many diseases, including cancer, diabetes, and rheumatoid arthritis, among others [27]. Several studies have been conducted on the antioxidant activity of bifidobacteria [5,27,28]. DPPH, hydroxyl, and superoxide radicals were used in this study to evaluate the antioxidant capacity of *B. longum* LBUX23. We observed that *B. longum* LBUX23 reduced DPPH radicals, mainly in intact cells following the cell debris and cell-free extract (Figure 5). In the case of hydroxyl radical scavenging capacity, it was shown that cell debris had major reductions in comparison with cell extract-free and intact cells, respectively, which suggests that it is not necessary that the bacteria be alive to have an antioxidant effect. In the case of superoxide anion radical scavenging capacity *B. longum* LBUX23 showed higher activity in cell-free extract, followed by intact cells and cell debris (Figure 5). Therefore, *B. longum* LBUX23 has a low effect over anion radical by itself. We showed a high anion radical effect in cell-free extract; therefore, it is suggested that this effect is due to a metabolite produced by the bacteria. *B. longum* LBUX23 did not contain coding genes related to catalase and superoxide dismutase. Additionally, it contains genes associated with ABC transporter ATP-binding protein, ferredoxin, thioredoxin, NADH oxidase, NADH peroxidase, and genes such as *nrdH* (PIB40_05210), *msrAB* (PIB40_04685), *PNPOx* (PIB40_09440), *AhpC* (PIB40_00775), *Bcp* (PIB40_00775), *trxA* (PIB40_02940), and *trxB* (PIB40_04850), which are related to such antioxidant activity. However, more research is needed on the expression of these genes and the mechanism that the produced proteins use to show the effect. This result is similar to *B. longum* LTBL16 [5], *B. Longum* BBMN68 [73], and *B. Longum* NCC2705 [74], which present different genes in response to ROS. In general, *Bifidobacterium longum* can decline the production of ROS, suppress oxidative stress, and reduce damage to the intestinal tract as a way of protecting intestinal epithelial cells [75]. Finally, *B. longum* LBUX23 is a candidate strain to test the antioxidant effect in an in vivo model.

## 5. Conclusions

In summary, the complete genome of *B. longum* LBUX23 is an effective tool for the discovery and identification of possible host-interaction and capacities of new probiotic strains. Among them, *B. longum* LBUX23 is grown in different carbon sources; apparently, it is considered a safe bacterium and shows adhesion ability, antimicrobial activity over emerging pathogens, tolerance of bile salts, high BSH activity, and antioxidant capacity. Our study indicates that *B. longum* LBUX23 has excellent probiotic properties, mainly in BSH activity and antioxidant capacity, and can be used as a candidate for antioxidant or anti-cholesterol effects in vivo models, among others. All the information generated in this study serves as evidence of the safety and functionality of this microorganism so that in the future, if any of these benefits are demonstrated in an in vivo model, *B. longum* LBUX23 could be used in biotechnological applications, human healthcare, and food science.

## Figures and Tables

**Figure 1 microorganisms-11-01648-f001:**
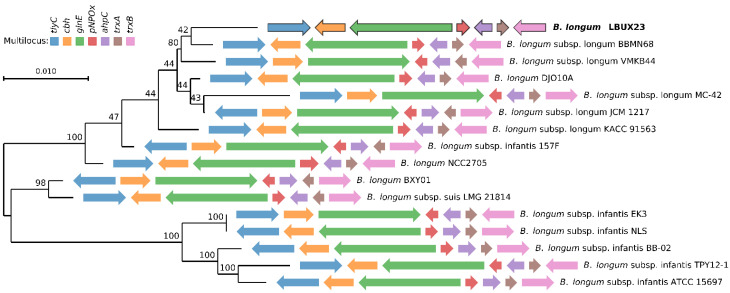
Multilocus sequence analysis (MLSA) and the maximum likelihood phylogenetic tree were constructed using genomic sequences of bifidobacterial strains obtained from the GenBank database. The *B. longum* LBUX23 strains identified and characterized in this study are indicated in bold font. The solid arrows represent the housekeeping genes utilized for constructing the multilocus (*tlyC*, *cbh*, *glnE*, *pNPOx*, *ahpC*, *trxA*, and *trxB*). The direction of the arrows reflects the orientation of these genes within the genome of each *Bifidobacterium* strain.

**Figure 2 microorganisms-11-01648-f002:**
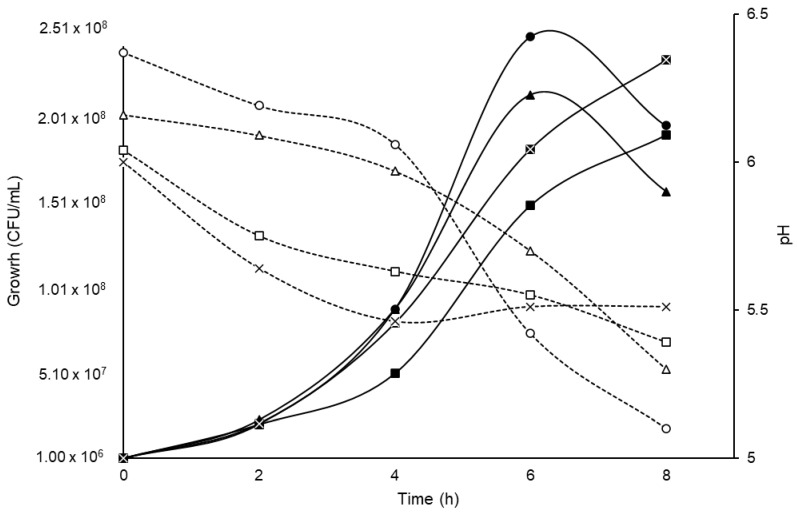
Effect of different carbon sources on the growth of *B. longum* LBUX23. Continuous lines in black correspond to the growth (CFU/mL). Black dotted lines indicate changes in pH over time regarding different carbon sources: 
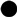
 sucrose, 
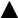
 FOS, 

 lactulose, and 
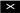
 lactose. In the case of pH: 
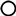
 sucrose, 
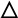
 FOS, 
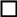
 lactulose, and 

 lactose.

**Figure 3 microorganisms-11-01648-f003:**
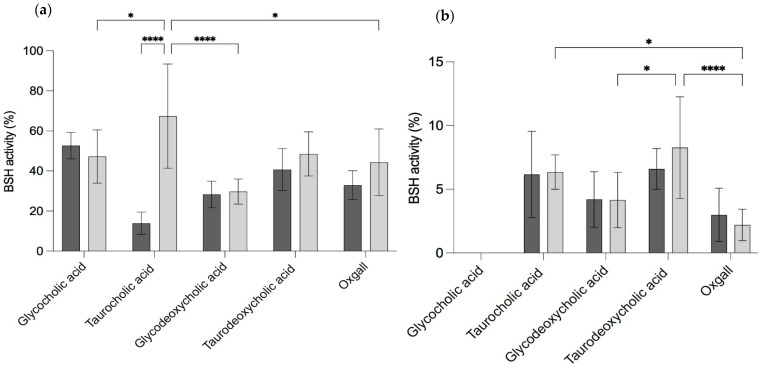
BSH activity (**a**) extracellular and (**b**) intracellular using glycol, and tauro cholic acids (primary bile salts), glycol, and tauro deoxycholic acids (secondary bile salts), and oxgall to 


*B. animalis* Bb-12 and 
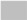

*B. longum* LBUX23. The ANOVA analysis **** means significant to *p <* 0.0001 and * to *p =* 0.01 between different *Bifidobacterium* species, while ns means no significant difference.

**Figure 4 microorganisms-11-01648-f004:**
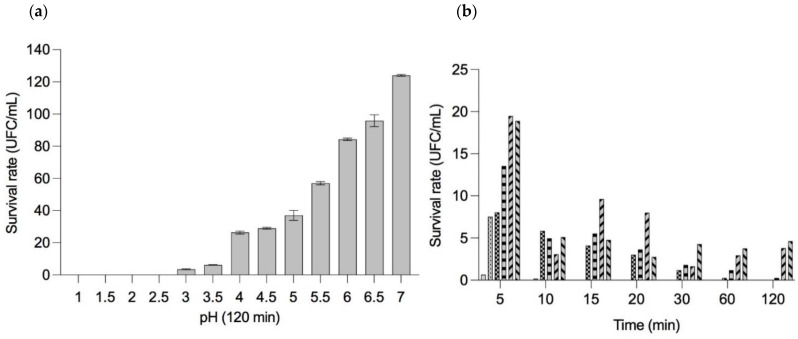
The survival rate of the percentage of survivors of *B. longum* LBUX23 (**a**) in different pH’s for 120 min (**b**) at different pH’s: 

 pH 1, 
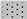
 pH 1.5, 
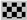
 pH 2, 
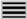
 pH 2.5, 
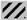
 pH 3, and 
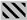
 pH 3.5.

**Figure 5 microorganisms-11-01648-f005:**
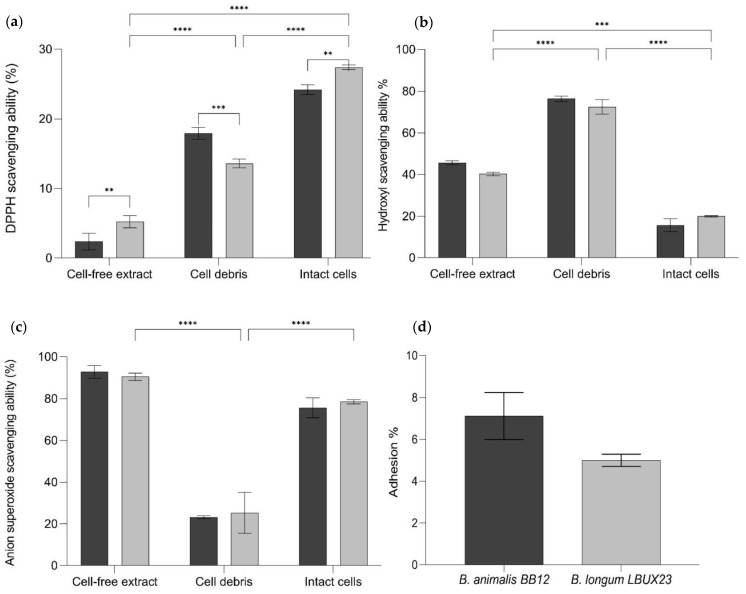
(**a**) DPPH scavenging ability (**b**) Hydroxyl scavenging ability (**c**) Anion superoxide scavenging ability in cell-free extract, cell debris, and intact cells. (**d**) Percentage of adhesion to Caco-2 cells. For all test to 


*B. animalis* Bb-12 and 


*B. longum* LBUX23. **** means significant *p <* 0.0001. The ANOVA analysis *** to *p =* 0.0003 and ** to 0.0095 between different *Bifidobacterium* species.

**Table 1 microorganisms-11-01648-t001:** Characterization of *B. longum* LBUX23.

	**(a) Antibiotic profile**											
	VA	AM	STX	GE	DC	CF	CLM	E	PE	TE	CFX	CPF
*B. animalis* Bb-12	S	R	S	S	R	R	S	S	S	R	S	R
*B. longum* LBUX23	S	R	S	S	R	S	S	S	S	S	S	S
	**(b) Hemolytic capacity**											
*B. animalis* Bb-12	Negative											
*B. longum* LBUX23	Negative											
	**(c) Antimicrobial activity**											
	*B. animalis* Bb-12						*B. longum* LBUX23					
*E. coli* ATCC 25922	++						+					
*E. coli* O157:H7	++						−					
*S. typhi* ATCC14028	++						−					
*S. aureus* ATCC 6538	+						−					
*K. pneumoniae* MDR	++						+					
*P. mirabiilis* MDR	++						+					
*P. aeruginosa* XDR	++						+					
	**(d) Bile salt tolerance assay**											
	GCA			TCA			GDCA			TDCA		OXG
*B. animalis* Bb-12	R			R			R			R		R
*B. longum* LBUX23	R			R			R			R		R

Where VA: vancomycin, AM: ampicillin, ST: trimethoprim-sulfamethoxazole, GE: gentamicin, DC: dicloxacillin, CF: cephalothin, CLM: Clindamycin, E: erythromycin, PE: penicillin, TE: tetracycline, CFX: cefotaxime, CPF: ciprofloxacin. S: Sensitive; R: resistant; +: weak inhibition; ++: strong; −: absence of inhibition. GCA: glycocholic acid; TCA: taurocholic acid; GDCA: glycodeoxycholic acid; TCDA: taurodeoxycholic acid and OXG: oxgall. All bile salts were from 0.1 to 0.5%.

**Table 2 microorganisms-11-01648-t002:** Genes associated with antioxidant activity identified in *B. longum* LBUX23.

Locus Tag	Start	Stop	Gene	Function
PIB40_05210	1206144	1206410	*nrdH*	Glutaredoxin-like protein NrdH
PIB40_04685	1074740	1075714	*msrAB*	Peptide methionine sulfoxide reductase msrA/msrB
PIB40_09440	2242860	2243264	*PNPOx*	Putative flavin-nucleotide-binding protein
PIB40_04855	1125603	1126166	*AhpC*	NADH-dependent peroxiredoxin subunit C
PIB40_00775	158528	159112	*Bcp*	Thioredoxin-dependent peroxiredoxin
PIB40_02940	670951	671325	*trxA*	Thioredoxin domain-containing protein
PIB40_04850	1123518	1125434	*trxB*	Thioredoxin reductase (NADPH)

## Data Availability

The data presented in this study are openly available from the NCBI in the BioProject PRJNA924960.

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
