# Peer review of "Bifidobacterium longum LBUX23 Isolated from Feces of a Newborn; Potential Probiotic Properties and Genomic Characterization"

_microorganisms, 2023, doi:10.3390/microorganisms11071648_

Round 1

Reviewer 1 Report

A manuscript with very interesting findings.  Just one comment: With regard to the antibiotic resistance profiling, do the authors have any comments about the presence of beta-lactam resistance affecting the generally regarded as safe (GRAS) status of strain LBUX23? The antibiotic resistance determinants may not be plasmid-borne, but wouldn't the chromosomal DNA act as donor DNA (in horizontal gene transfer) upon lysis of the bacterium?

The manuscript is generally well-written except for quite a few grammatical errors and lack of attention to detail. Examples include:

1. Title - Bifidobacterium longum is italicized.

2. Abstract, line 31 - "...a neonate's feces..." - didn't LBUX23 come from a single neonate's fecal material?

3. Abstract, lines 33 and 39 - define CDS and BSH on first use.

4. Line 54 - species - it's species in singular and plural forms.

5. Lines 55-56 - this sentence is incorrect - should be subsp. longum, subsp. infantis and subsp. suis. (the word ssp. is the plural for species.)

6. Lines 125-126 - 16S rRNA gene sequencing. There is no such thing as 16S rDNA - you are sequencing the gene (which is DNA).

7. Lines 130 and 132 - (MLSA) and MLSA, respectively.

8. Line 272 - "...we isolated a Gram-positive bacillus..." or "we isolated a Gram-positive rod-shaped bacterium...". (The word bacillus is given to rod-shaped bacteria.)

9. Lines 273 and 276 - BLAST and GenBank, respectively.

10. Table 1 footnote - S, Sensitive (not Sensible).

11. Line 349 - BSH, not BHS.

12. Line 369 - "At pH 2.5, ..." (not To).

The authors are encouraged to go through their manuscript again to tidy up the errors listed above and others they might find.

Author Response

Revisor 1

-A manuscript with very interesting findings.  Just one comment: With regard to the antibiotic resistance profiling, do the authors have any comments about the presence of beta-lactam resistance affecting the generally regarded as safe (GRAS) status of strain LBUX23? The antibiotic resistance determinants may not be plasmid-borne, but wouldn't the chromosomal DNA act as donor DNA (in horizontal gene transfer) upon lysis of the bacterium?

Answer

The point of the revisor were discussed in lines 521-524.

“However, it exists the possibility that B. longum LBUX23 and other generally recognized as safety (GRAS) bacteria could transfer the genomic material by transformation mechanism, therefore it is necessary to define the GRAS status again from a molecular point of view”.

-Comments on the Quality of English Language

The manuscript is generally well-written except for quite a few grammatical errors and lack of attention to detail. Examples include:

  1. Title - Bifidobacterium longum is italicized.

Answer: Bifidobacterium longum were italicized

  1. Abstract, line 31 - "...a neonate's feces..." - didn't LBUX23 come from a single neonate's fecal material?

Answer: This line was modified, please see the line

  1. Abstract, lines 33 and 39 - define CDS and BSH on first use.

Answer: CDS and BSH were defined when first use

  1. Line 54 - species - it's species in singular and plural forms.

Answer: It was modified. Please see line 58-59

  1. Lines 55-56 - this sentence is incorrect - should be subsp. longum, subsp. infantis and suis. (the word ssp. is the plural for species.)

Answer: It was modified. Please see line 61.

  1. Lines 125-126 - 16S rRNA gene sequencing. There is no such thing as 16S rDNA - you are sequencing the gene (which is DNA).

Answer: The mistake was corrected. Please see line 131.

  1. Lines 130 and 132 - (MLSA) and MLSA, respectively.

Answer: The mistake was corrected. Please see line 135-137.

  1. Line 272 - "...we isolated a Gram-positive bacillus..." or "we isolated a Gram-positive rod-shaped bacterium...". (The word bacillus is given to rod-shaped bacteria.)

Answer: The mistake was corrected. Please see line 279.

  1. Lines 273 and 276 - BLAST and GenBank, respectively.

Answer: The mistake was corrected. Please see lines 280 and 283.

  1. Table 1 footnote - S, Sensitive (not Sensible).

Answer: Footnote was modified. It was used sensitive instead of sensible

  1. Line 349 - BSH, not BHS.

Answer: The mistake was corrected. Please see line 357.

  1. Line 369 - "At pH 2.5, ..." (not To).

Answer: The mistake was corrected. Please see line 376

The authors are encouraged to go through their manuscript again to tidy up the errors listed above and others they might find.

Answer:

The entire document was reviewed by a native colleague.

Reviewer 2 Report

Dear authors:

In the manuscript entitled “Bifidobacterium longum LBUX23 Isolated from Feces of a New-Born; Potential Probiotic Properties and Genomic Characterization” by Reyes-Castillo et al., the authors report the findings of the characters of probiotic potential B. longum LBUX23, of a strain isolated from neonates’ feces. Based on the biochemical and genomic functionality of B. longum LBUX23, it showed the potential probiotic properties especially bile salt hydrolase activity and antioxidant capacity. The work is quite intriguing and sheds light on how B. longum LBUX23 might be used to improve human health. The suggestions for specifics were as follows:

1 Page 1, Line 28-Page 2, 49

The impacts of various carbon sources on the growth of Bifidobacterium longum LBUX23 and the findings of adhesion assays of B. longum LBUX23 are not mentioned in the abstract of the manuscript.

2 Page 1, Line 36

The word "BSH" should be written in its entire name when it first appears.

3 Page 8, Line 337

Please explain in the annotation what these antibiotics' abbreviations mean in the table 1a.

Besides, “c) Bile salt tolerance assay” should be “d) Bile salt tolerance assay”.

4) Page 10, Line 370-373

The author mentioned “The genomic analysis of B. longum 370 LBUX23 showed the presence of argC, argH, and dapA genes associated with high acid tolerance”. Please provide more details on how this conclusion was reached.

5) Page 11, Line 383-387; Page 13, Line 420-425

Please describe more in details how these genes are linked to the antioxidant activity and adhesion assay of B. longum 370 LBUX23.

I would advise the authors to have a native English speaker to check the language.

Author Response

In the manuscript entitled “Bifidobacterium longum LBUX23 Isolated from Feces of a New-Born; Potential Probiotic Properties and Genomic Characterization” by Reyes-Castillo et al., the authors report the findings of the characters of probiotic potential B. longum LBUX23, of a strain isolated from neonates’ feces. Based on the biochemical and genomic functionality of B. longum LBUX23, it showed the potential probiotic properties especially bile salt hydrolase activity and antioxidant capacity. The work is quite intriguing and sheds light on how B. longum LBUX23 might be used to improve human health. The suggestions for specifics were as follows:

-1 Page 1, Line 28-Page 2, 49

The impacts of various carbon sources on the growth of Bifidobacterium longum LBUX23 and the findings of adhesion assays of B. longum LBUX23 are not mentioned in the abstract of the manuscript.

Answer:

Abstract was modified including information about findings of carbon sources and adhesion assay. Please see the abstract.

-2 Page 1, Line 36

The word "BSH" should be written in its entire name when it first appears.

Answer:

It was written in entire name when if first appeared, please see abstract.

-3 Page 8, Line 337

Please explain in the annotation what these antibiotics' abbreviations mean in the table 1a.

Besides, “c) Bile salt tolerance assay” should be “d) Bile salt tolerance assay”.

Answer:

The abbreviation of antibiotics and the complete name of them were included at the end of table 1.

It was used d)Bile salt tolerance assay instead of c)Bile salt tolerance assay

-4 Page 10, Line 370-373

The author mentioned “The genomic analysis of B. longum 370 LBUX23 showed the presence of argCargH, and dapA genes associated with high acid tolerance”. Please provide more details on how this conclusion was reached.

Answer:

In results section the following comment was included:

“, which have been reported as responsible of the ability to tolerate acid environments in other species of B. longum”

The details about how this conclusion was reached are included in discussion section please see:

“In the case of pH 3 and 3.5, B. longum LBUX23 is capable of surviving < 10%, probably due to the presence of argC (PIB40_02085), argH (PIB40_02045), and dapA (PIB40_02760) genes. These genes have been previously reported by Sundararaman et al. 2021 [26] as those who confer to B. longum NCIM 5672 a survival advantage in an acid environment (pH 3). In the case of B. Longum LBUX23 those genes could be responsible about the acid tolerance, nevertheless more research is needed to prove their expresiona and to elucidate how the proteins codified by those genes act against acidity”.

-5 Page 11, Line 383-387; Page 13, Line 420-425

Please describe more in details how these genes are linked to the antioxidant activity and adhesion assay of B. longum 370 LBUX23.

Answer:

Details about the genes related to antioxidant and adhesion were included in results and discussion section.
